# Uncommon Presentation of Sarcoidosis with Severe Thrombocytopenia and Hemorrhagic Diathesis

**Dorela Lame, Michelangelo Pianelli** , **Shahram Kordasti, Erika Morsia, Attilio Olivieri and Antonella Poloni** *

AOU Delle Marche, Università Politecnica Delle Marche, 60126 Ancona, Italy; dorela.lame@gmail.com (D.L.)
* Correspondence: a.poloni@univpm.it

**Abstract:** Sarcoidosis, a multi-organ system disease, often presents insidiously. Thrombocytopenia in sarcoidosis is frequent because of hypersplenism, granulomas infiltrating the bone marrow, or immune thrombocytopenia (ITP). The diagnosis of ITP relies on exclusionary criteria, given the absence of a definitive laboratory diagnostic feature. In the era prior to modern ITP management, sarcoidosis-associated ITP was known to manifest severely, often showing resistance to treatment and an increased risk of mortality. In this case, we present a young male who was admitted to a district hospital's emergency room, displaying symptoms of hematuria, gingival bleeding, and a petechial rash. Blood tests revealed severe thrombocytopenia with a platelet count of 0, while all other metabolic and serological exams returned normal results. Infectious and autoimmune causes were ruled out, and a bone marrow examination excluded any hematological disorder. Initial management, including platelet transfusion and presumptive treatment for ITP with dexamethasone and Human Immunoglobulin IV (IVIG), failed to improve the patient's platelet count or alleviate the hemorrhagic diathesis. Second-line therapy with Rituximab and Methylprednisolone was initiated with no benefit. Considering the hemorrhagic signs and the delayed response of Rituximab, we shifted to third-line therapy with Romiplostim at the maximal dose and continued Methylprednisolone. The platelet count recovered completely after the second Romiplostim administration (over $350 \times 10^9$ platelets/L) and Methylprednisolone was rapidly tapered. To further study the causes of thrombocytopenia a total body CT scan was performed and it identified non-homogeneously hypodense tissue in the bilateral hilar area extending medially to the subcarinal area, suggesting possible lymphatic origin and raising suspicion of sarcoidosis. Further investigations, including Angiotensin Converting Enzyme (ACE) titration, bronchoscopy, bronchoalveolar lavage, and EndoBronchial UltraSound-guided TransBronchial Needle Aspiration (EBUS-TBNA), confirmed the diagnosis of sarcoidosis. Despite a mild restrictive insufficiency noted in spirometry, the patient remained asymptomatic with only a mild respiratory insufficiency, and hence, was enlisted for follow-up. As for the ITP, the platelet count remained normal over a year. Notably, while sarcoidosis onset often predates ITP onset by an average of 48 months, in our case the onset of the two diseases was simultaneously. Our case adds valuable information to the limited body of knowledge regarding the treatment of sarcoidosis-associated ITP.

**Keywords:** bleeding; ITP; thrombocytopenia; sarcoidosis; romiplostim

## 1. Introduction

Sarcoidosis is a multi-organ system disease of unknown origin characterized by the development of non-necrotizing granulomas in various organs. While it can affect individuals of all ethnic backgrounds, it is more commonly observed in African Americans and Scandinavians. Sarcoidosis can manifest at any age, but it is most frequently diagnosed in adults aged between 30 and 50 years old.

The condition can impact any organ, with intrathoracic involvement being present in approximately 90% of patients. This typically involves symmetrical bilateral hilar adenopathy and/or diffuse lung micronodules, primarily along the lymphatic structures, which are

the most affected system. In addition to pulmonary manifestations, sarcoidosis can lead to extrapulmonary symptoms such as skin lesions, uveitis, liver or splenic involvement, peripheral and abdominal lymphadenopathy, and peripheral arthritis. These extrapulmonary manifestations are observed in 25–50% of cases [1]

Diagnosing sarcoidosis is not standardized but relies on three main criteria: a clinical and/or radiological presentation consistent with the disease, histological evidence of non-necrotizing granulomatous inflammation in one or more tissues, and the exclusion of alternative causes of granulomatous disease [1].

Sarcoidosis is frequently linked with hematological abnormalities like anemia and leukopenia, the presentation of isolated thrombocytopenia is relatively uncommon.

The first incidence of thrombocytopenia in sarcoidosis was documented by Jersild et al. in 1938, with further connections primarily identified in retrospective studies, exhibiting an incidence rate of 1–2% [2]

The causes of thrombocytopenia in sarcoidosis can be attributed to three primary mechanisms, with many patients potentially exhibiting a combination of these mechanisms [3,4]. The first proposed pathophysiological mechanism of thrombocytopenia in sarcoidosis is linked to hypersplenism and splenomegaly. According to Fordice et al. [5], the prevalence of splenomegaly was estimated at 10% based on a study that examined 6074 cases of sarcoidosis from 29 publications. Sequestration of platelets in the spleen results in their destruction. In some instances, splenomegaly may be associated with hypersplenism secondary to portal hypertension, leading to peripheral pancytopenia. Furthermore, moderate splenomegaly is also consistent with an autoimmune mechanism, which results in the peripheral destruction of platelets. Therefore, in patients with splenomegaly and varying degrees of cytopenia (leukopenia, anemia, thrombocytopenia), thrombocytopenia related to hypersplenism in sarcoidosis should always be considered.

The second mechanism involves bone marrow involvement. Although bone marrow granulomatous infiltration in sarcoidosis appears to be rare and not clearly linked to thrombocytopenia, it remains a consideration.

The third mechanism is autoimmunity. Sarcoidosis has been associated with various immunological abnormalities, including a T helper 1-driven granulomatous response and an active humoral immune reaction. Autoimmune conditions like hemolytic anemia and immune thrombocytopenia (ITP) have been documented in sarcoidosis cases. In 1985, Lawrence identified elevated platelet-associated IgG levels in a patient with both thrombocytopenia and sarcoidosis, suggesting the possibility of antibody-mediated platelet destruction [6]. Numerous immunological irregularities have been described in sarcoidosis, with peripheral lymphopenia contrasting with the recruitment of T helper cells at sites of active disease. The proliferation of B cells, stimulated by activated T cells, is reflected in the characteristic polyclonal hypergammaglobulinemia observed in sarcoidosis.

Immune thrombocytopenia (ITP) is defined as isolated thrombocytopenia with platelet count below $100 \times 10^9$/L on at least two occasions in the absence of drug-induced thrombocytopenia, hypogammaglobulinemia, suggesting a underlying common variable immunodeficiency (CVI), infections, splenomegaly, portal hypertension, or pancytopenia [7]. As of now, ITP remains a diagnosis of exclusion due to the absence of a definitive laboratory diagnostic feature. Before modern ITP management, sarcoidosis-associated ITP was observed to present severely with a tendency towards treatment unresponsiveness and increased mortality [8]. We report a case of sarcoidosis, presenting with severe thrombocytopenia and hemorrhagic diathesis treated successfully with high dose of Romiplostim.

## 2. Detailed Case Presentation

In August 2022, a 34-year-old Asian male (ECOG PS 1) was admitted to a district hospital's emergency room, presenting with hematuria, gingival bleeding, and a petechial rash. The patient had unremarkable health history and no prescribed medication, no smoking nor drinking history; a familial anamnesis was negative for bleeding disorders and hematological diseases.

The patient's initial blood tests revealed a complete absence of platelets, registering at 0. Hemoglobin levels were measured at 13.4 g/dL, white blood cell count at $8.5 \times 10^6$/L, and coagulation markers fell within the normal range. The peripheral blood smear displayed no schistocytes, no indications of hematologic disorders, and no visible platelets. Additionally, the proteinogram and serum electrolytes, lactate dehydrogenase, D-dimer levels, renal function, liver function, and coagulation tests all yielded normal results. Further investigations included negative serology for HCV, HIV, HBV, Parvovirus, COVID-19 antigen, negative EBV-DNA and CMV-DNA, and a negative H pylori antigen test in feces. ANA, ENA, ANCA, and anti-phospholipid antibodies all fell within their respective normal ranges. Both a cerebral CT scan and chest radiography returned negative findings.

Initially, the patient received a platelet transfusion, which failed to elicit any response in terms of increasing the platelet count or alleviating hemorrhagic symptoms. Given the lack of definitive findings from the screening tests and the unresponsiveness to platelet transfusions, a presumptive treatment for ITP was initiated. This treatment included Dexamethasone at a dose of 40 mg for 4 days and Human Immunoglobulin IV (IVIG) at a dosage of 2 g/kg (1 g/kg for 2 days). Despite these interventions, the platelet count remained consistently below $5 \times 10^9$/L, and the hemorrhagic diathesis showed no signs of improvement.

Consequently, it became necessary to initiate a second-line therapy, which involved administering Rituximab at a dosage of 375 mg/m$^2$ and Methylprednisolone at a dose of 1 mg/kg. However, two days later, there was no discernible change in the platelet count, leading to the patient's transfer to our clinic.

Upon admission, the patient's platelet count remained at $1 \times 10^9$/L, while the hemoglobin level measured 10.4 g/dL and white blood cell count was $15 \times 10^6$/L, with all other metabolic and serological parameters falling within normal ranges. Examination of the peripheral blood smear revealed an absence of platelets, and the red blood cells exhibited a moderate degree of anisopoikilocytosis. No schistocytes or immature cells were observed. Despite experiencing symptoms such as hematuria, a petechial rash, and gingival bleeding, there were no new signs of complications. However, the patient reported weakness and palpitations due to anemia. All previously conducted screening tests were repeated and yielded results consistent with previous findings. There were no indications of infection, and the physical examination did not reveal any abnormalities.

Considering the presence of hemorrhagic symptoms, the correlation with anemia, and the delayed response to Rituximab as documented in the literature, we transitioned to third-line therapy. This involved administering Romiplostim at a dosage of 9 ug/kg alongside continued Methylprednisolone at a dose of 1 mg/kg, with a complete response on the platelet count as described below.

To further investigate the thrombocytopenia, the patient underwent a series of bone marrow (BM) examinations:

(i)  BM aspirate revealed normal hematopoiesis with hyperplastic megakaryopoiesis, showing no morphological signs of a hematological disorder or lymphoid infiltration.
(ii)  Cytogenetic analysis demonstrated a normal karyotype (46, XY).
(iii)  Flow cytometry analysis did not detect any abnormalities.
(iv)  BM biopsy indicated a cellularity of 50%, with normal maturation of myeloid and erythroid lineages. Megakaryocytes were abundant, primarily small in size.

Furthermore, a comprehensive total body CT scan was conducted, which ruled out the presence of adenopathy and adenomegaly. However, within the thoracic region, the scan identified several millimetric parenchymal nodules. These nodules, due to their small size, proved challenging to characterize fully. The largest among them had a maximum diameter of 0.81 cm and was located in the middle lobe. Additionally, there was non-homogeneously hypodense tissue observed in the bilateral hilar area, extending medially to the subcarinal area. This finding raised suspicion of a lymphatic origin and prompted consideration of sarcoidosis as a potential diagnosis (see Figure 1).

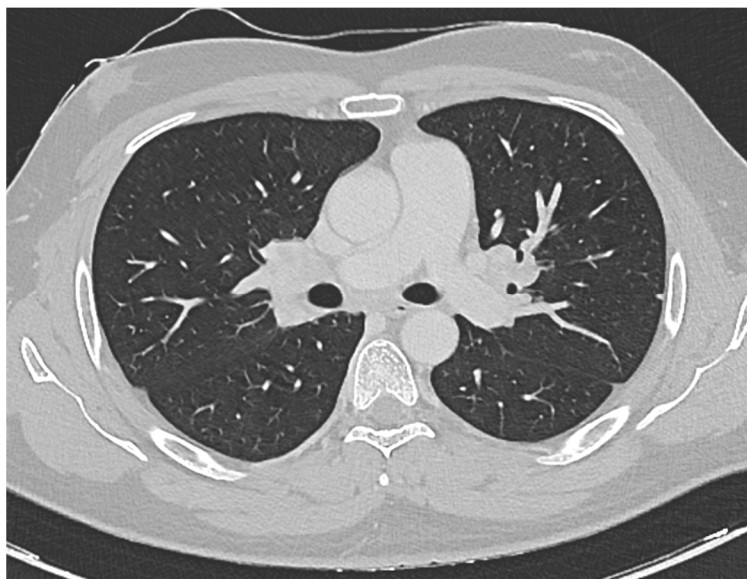

**Figure 1.** CT scan showing the presence of non-homogeneously hypodense tissue in the bilateral hilar area that extends medially up to the subcarinal area, likely of lymphatic origin worthy of cyto-histological evaluation corresponding to Sarcoidosis.

Further investigations were pursued, which included:

I.      Angiotensin Converting Enzyme (ACE) levels, which were measured at 47 U/L.

II.      Bronchoalveolar lavage, revealing a cellularity of 88,500 cells/mL, with a composition of 10% neutrophils, 60% macrophages, and 10% lymphocytes. The CD4/CD8 ratio was notably elevated, exceeding 3.6.

III.      EndoBronchial UltraSound-guided TransBronchial Needle Aspiration (EBUS-TBNA) of a hilar lymph node, which revealed the presence of aggregates of histiocytic cells without necrosis. This was confirmed by excluding infectious agents through Ziehl-Neelsen staining, ultimately classifying it as a non-necrotizing granuloma consistent with sarcoidosis.

Following a thorough pneumological evaluation, the diagnosis of sarcoidosis was conclusively confirmed.

Although spirometry indicated a mild restrictive insufficiency with FVC at 82% and TLC at 75%, the patient remained asymptomatic, experiencing only mild respiratory insufficiency. Consequently, the patient was scheduled for regular follow-up appointments.

Following the initial administration of Romiplostim, a thrombopoietin receptor agonist, there was a gradual and notable increase in the patient's platelet count observed over a span of seven days, ultimately reaching $33 \times 10^9$/L. The immature platelet fraction (IPF) concurrently registered at 43%. In response to this positive trend, a subsequent dose of Romiplostim was judiciously administered, and concurrently, the dosage of Methylprednisolone was promptly reduced.

Within the subsequent week, a remarkable improvement in the platelet count was observed, soaring to $356 \times 10^9$/L. Importantly, the hemorrhagic symptoms that had previously afflicted the patient were completely ameliorated. Noteworthy, however, is the temporal association between the full recovery of the platelet count and the onset of exacerbated symptoms, including headaches, photophobia, nausea, walking instability, and dizziness. These symptoms, notably, are not recognized as common side effects of Romiplostim. It is crucial to highlight that, despite these emerging symptoms, a cerebral thromboembolism attributed to Romiplostim was meticulously ruled out through a cerebral CT scan. Subsequently, appropriate and tailored supportive therapy was administered to effectively manage and alleviate the aforementioned symptoms.

In the subsequent blood tests conducted a week later, the platelet count measured $258 \times 10^9$/L, hemoglobin levels were at 11.4 g/dL, and white blood cell count was $9.8 \times 10^6$/L. Temporarily suspending Romiplostim, we continued to monitor the platelet count, which consistently remained above $200 \times 10^9$/L (as clearly depicted in Figure 2). Furthermore, the anemia resolved following the resolution of bleeding.

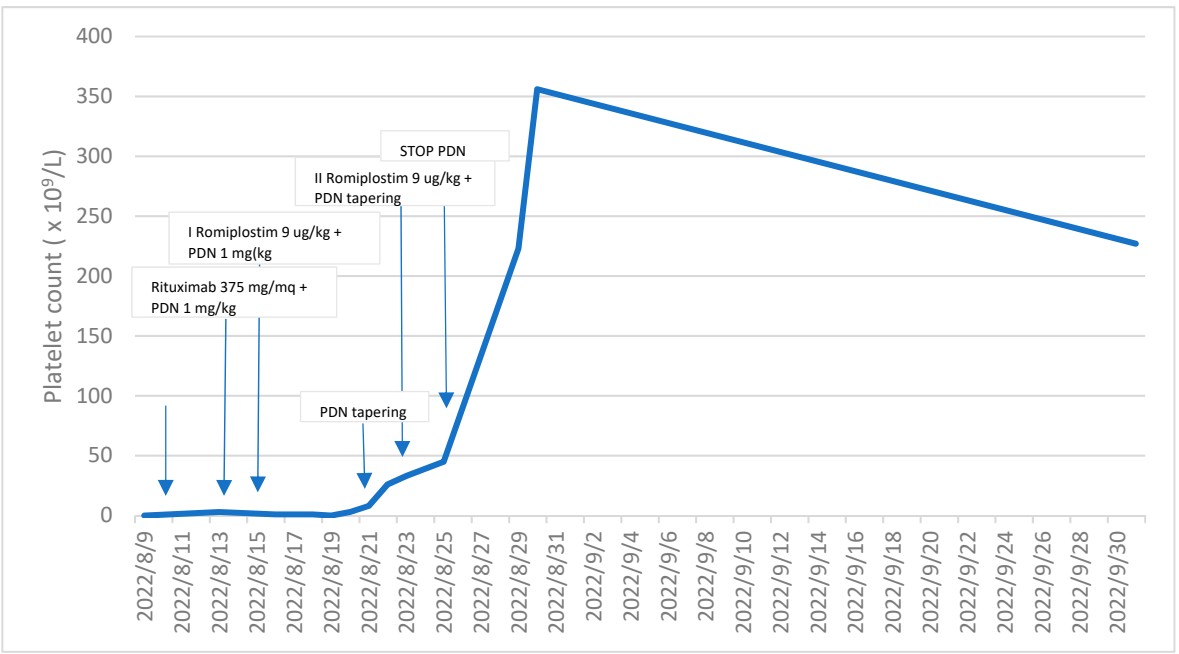

**Figure 2.** This chart illustrates the different treatment stages and their impact on the platelet count throughout the course of treatment. As of the latest update in July 2023, the platelet count continues to be consistently above $150 \times 10^9$/L.

Regular outpatient visits were scheduled every three months, during which the patient maintained excellent health conditions. All test results remained within the normal range, and the patient continued to lead a normal lifestyle.

## 3. Discussion

Sarcoidosis, characterized by granulomatous inflammation, exhibits a notable prevalence among African American and Scandinavian populations. However, it is noteworthy that our patient is of Asian descent, a demographic with a traditionally lower incidence of sarcoidosis [1]. Our case highlights the rare simultaneous onset of sarcoidosis and ITP, as sarcoidosis typically precedes ITP by an average of 48 months. In a study conducted by Mahévas et al. in France, the average time between the onset of sarcoidosis and the onset of ITP ranged from 6 to 216 months [4]. However, among the 20 patients in the study, five had a simultaneous presentation of ITP and sarcoidosis. Mahévas et al. demonstrated that among patients with both ITP and sarcoidosis, 60% had either the onset or a relapse of sarcoidosis at the same time as the onset of ITP. Additionally, 66% of these patients had thoracic involvement. Interestingly, there were no discernible differences in the clinical presentation of sarcoidosis among patients in whom sarcoidosis preceded, occurred simultaneously with, or followed ITP.

Treatment approaches varied among the patients in the study. One patient, who was already receiving oral corticosteroid therapy for sarcoidosis, did not require specific therapy for ITP but needed an increase in corticosteroid dosage. Nineteen out of twenty patients received specific treatment for ITP, with seven of them requiring second- or third-line therapies.

While TPO-R agonists like Eltrombopag, Romiplostim, and Avatrombopag are used as second-line therapy for primary ITP [7], there is limited evidence and no explicit guidelines for secondary ITP.

In our case, the patient did not respond to IgIV+Dexamethasone and Rituximab administered concurrently with Methylprednisolone. However, the patient showed a very positive response to just two injections of Romiplostim for ITP, and the sarcoidosis did not necessitate specific therapy (possibly due to the Rituximab treatment and an extended period of steroid use). The true extent of sarcoidosis before these therapies cannot be definitively determined.

While secondary immune thrombocytopenia (ITP) is considered a rare condition, the presented case holds significance as it exemplifies the coincidental manifestation of these diseases in a single patient Our case study contributes valuable insights to the limited body of knowledge concerning the treatment of sarcoidosis-associated ITP [9,10]

## 4. Conclusions

Although secondary immune thrombocytopenia (ITP) is acknowledged as a rare medical condition, the case presented assumes heightened significance by exemplifying the fortuitous co-occurrence of these diseases within a singular patient. Our comprehensive case study not only sheds light on this infrequent association but also serves to contribute valuable insights to the limited body of knowledge regarding the treatment strategies for sarcoidosis-associated ITP

While no established guidelines specifically address the management of immune thrombocytopenia (ITP) in the context of sarcoidosis, it is deemed reasonable, particularly in emergency situations involving significant bleeding, to adhere to the prevailing guidelines recommended for the treatment of primary immune thrombocytopenia.

**Author Contributions:** Conceptualisation A.P. and D.L., formal analysis D.L. e M.P., writing original draft D.L.; writing, review and editing D.L., M.P., E.M., S.K., A.O., A.P., supervision A.P. All authors have read and agreed to the published version of the manuscript.

**Funding:** This research recoeved no external funding.

**Institutional Review Board Statement:** The study was conducted in accordance with the Declaration of Helsinki. On 30/08/2022 the patient was enrolled in the Italian Registry of ITP, number "NCT03465020".

**Informed Consent Statement:** Informed consent was obtained from all subjects involved in the study.

**Data Availability Statement:** Not applicable.

**Conflicts of Interest:** The authors declare no conflict of interest.

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
