# Peer review of "Uncommon Presentation of Sarcoidosis with Severe Thrombocytopenia and Hemorrhagic Diathesis"

_hematolrep, doi:10.3390/hematolrep16010013_

Round 1

Reviewer 1 Report

Comments and Suggestions for Authors

Sarcoidosis can affect disease through direct tissue damage from either direct involvement by granulomatous inflammation, indirectly through the actions of inflammatory cytokines produced as part of the inflammatory process, or by disturbing other disease pathways (such as increasing the conversion of inactive vitamin D-25OH into its active form vitamin D-1,25OH). In the absence of any of these 3 statements, then the only conclusion is that the patient is experiencing two uncommon disorders.

Furthermore, when a patient has both sarcoidosis and another uncommon medical condition, it is often challenging to determine if sarcoidosis and the other uncommon medical condition are related to each other. As a rule, disease manifestations that are due to active sarcoidosis inflammation should respond appropriately to corticosteroids, which remain the cornerstone for treating sarcoidosis. For example, if a patient with sarcoidosis experiences frequent infections and is found to have hypogammaglobulinemia, the hypogammaglobulinemia is unlikely to respond to treatment with corticosteroids for sarcoidosis and thus represents a separate disease process needing treatment appropriate for hypogammaglobulinemia, often IVIG. Another example is when a patient with sarcoidosis experiences kidney stones and later found to have hyperparathyroidism (rather than enhanced conversion of vitamin D-1,25OH), this will not respond to treatment of sarcoidosis and will require parathyroid resection.

This case report tackles a longstanding observation of sarcoidosis and ITP occurring in the same patient. The concerns here include (1) The sarcoidosis diagnosis was based on a lymph node needle biopsy showing aggregates of histiocytes, which cannot exclude the possibility of reactive lymphadenopathy, and the patient dose not appear to have any other manifestation of sarcoidosis other than trying to find a cause for their ITP.  It may be argued that in this case the patient had 2 identified treatable conditions (sarcoidosis, ITP) to base an initial approach to treatment that additional testing was unnecessary, but if instead the major concern was that this patient possibly had a cancer, then certainly additional testing (PET/CT scan, surgical biopsy / mediastinoscopy) would be performed to obtain additional evidence to support a diagnosis of sarcoidosis. (2) The ITP did not respond to treatment for sarcoidosis using corticosteroids, raising a concern that the ITP in this case was not due to active sarcoidosis inflammation. (3) The ITP responded only after using romiplostim, an established treatment for ITP. https://www.ncbi.nlm.nih.gov/pmc/articles/PMC4304598/

Therefore, I would recommend that the manuscript be carefully revised to de-emphasize that sarcoidosis is a cause of ITP, rather that these two conditions have been observed to co-exist in some patients and treatments appropriate for ITP must be considered if the thrombocytopenia does not respond to treatment for sarcoidosis.

Author Response

Clarifications and Considerations on the Patient's Case

Dear reviewer,

Thanking you for your comments, we would like to share our perspectives on the following concerns:

  1. The diagnosis of ITP is essentially an exclusion diagnosis, with the response to steroids serving as an adjuvant criterion in a patient that doesn’t respond to platelet trasfusions and has haemorrhagic symptoms. In our patient's case, unresponsiveness to both corticosteroids and intravenous immunoglobulin prompted for a secondary ITP, as per ASH guidelines, in cases where the standard approaches prove ineffective, an exploration of alternative causes for ITP is recommended. Literature suggests an association between ITP and sarcoidosis, and in some instances, ITP precedes the onset of sarcoidosis or is concomitant to its diagnosis, although the pathophysiology remains unclear. Given the patient's young age, the absence of risk factors and a negative total body CT scan, a search for possible cancer was not pursued.
  2. The hypothesis of sarcoidosis was put forth by our radiologists based on a CT scan highly suggestive of sarcoidosis. Further supporting this diagnosis were findings from a lymph node needle biopsy, the CD4/CD8 ratio in bronchoalveolar lavage, and the presence of mild restrictive pulmonary insufficiency.
  3.  The patient exhibited a positive response to romiplostim, albeit its use was considered "off-label" as TPO-RA administration typically commences three months post-diagnosis.

Reviewer 2 Report

Comments and Suggestions for Authors

Dear Authors,

It is an honor for me to review this paper. This report presents an interesting case of simultaneous bleeding manifestations caused by ITP, followed by a confirmatory diagnosis of sarcoidosis in the lung. However, there are some areas for improvement that could enhance the paper.

Comment 1: In lines 38-39, it is mentioned that sarcoidosis commonly occurs in African American and Scandinavian populations. However, this patient is of Asian descent. Could you please elaborate on the risk percentage of this disease in Asian populations? Kindly include this information in the discussion section.

Comment 2: Although this patient has minimal to no pulmonary involvement, could you please discuss what should be prioritized during their follow-up? Additionally, consider addressing whether the patient needs long-term treatment for sarcoidosis and/or secondary ITP in the discussion section.

Comment 3: It is appropriate to include in the conclusion that while secondary ITP is a rare condition, primarily caused by the presence of sarcoidosis, this case is noteworthy as it demonstrates the coincidental occurrence of these diseases in a patient. This finding should be taken into consideration in future patient evaluations.

Comments on the Quality of English Language

The quality of English writing is adequate, some minor changes may be required.

Author Response

Dear reviewer,

We deeply value your comments and suggestions.

Regarding comment 1: the incidence in the Asian populations is  1-2 per 100,000 population, we will add this information in the discussion.

Regarding comment 2: The patient is enrolled in a hematological and pulmonary follow-up program. In this particular case, the patient remains asymptomatic for sarcoidosis, notwithstanding the fact that the treatment administered for sarcoidosis involves immunosuppression, such as corticosteroids and rituximab—both of which were administered to address the ITP. Consequently, the true extent of sarcoidosis involvement prior to these therapies remains uncertain. Nevertheless, over the course of a one-year follow-up, there have been no discernible manifestations of sarcoidosis in the patient nor haemorragic symptoms.

Regarding comment 3: we will emphasase this aspect in the conclusion. 

Reviewer 3 Report

Comments and Suggestions for Authors

This case report is well-written and presented. The introduction provides a complete and very interesting field analysis. However, it could be improved by discussing the possible disease causes, and the possible common factors for the higher prevalence in Scandinavia and in African Americans. Sarcoidosis has probably at least an auto-immune component, that explains the long disease course, or its potential chronicity. Although the pathological cause is not well-understood, it should be worth to discuss it in introduction. This case is also characteristic of the dramatic thrombocytopenia, along with mild sarcoidosis symptoms., and the possible simultaneous onset of ITP with the disease. It is striking to note the good responsiveness to the TPO-receptor agonist, Romiplostim. This drug only contains 4 replicates of 14 out of the 332 amino acids of thrombopoietin, linked to a Fc moiety. Could this suggest the presence of autoantibodies to Thrombopoietin, to epitopes different from the 14 amino acid sequence? Were they tested? The good recovery with Romiplostim is likely to exclude presence of autoantibodies to platelet glycoproteins.

Concerning minor comments:
The English language can be improved in abstract. 
To remain homogenous with the first 2 therapeutic lines, it should be useful to indicate at line 177 the patient's response to Romiplostim, although it is well described at line 222 and beyond, and on figure 2.
In legend to figure 1, it should be "corresponding" (and not "corrisponding").
On lines 226-227, it should be useful to mention if these therapy side effects are usual, and how they were overcome.
The legend to figure 2 must start with "This" (and not "is").
In the discussion section, it should be useful that the authors comment the causes for the "apparent" short time or concomitance between the onset of sarcoidosis and the severe thrombocytopenia.

Comments on the Quality of English Language

This case-report is globally acceptable, well presented, and only some minor corrections are necessary in the text, but a more extensive review of the abstract is suggested.

Author Response

Dear Reviewer,

First and foremost, thank you for the review and assistance in enhancing this work.

We have addressed the grammatical errors and improved the abstract as per your suggestions. However, we did not delve into the reasons behind the higher incidence of sarcoidosis in Scandinavians and Africans. Upon an initial literature review, we found incomplete data, and it falls outside the scope of this article, which focuses on the description of a clinical case.

Lastly, in our hospital, we do not have the possibility to investigate the presence of anti-thrombopoietin antibodies. Therefore, we have not conducted studies on the existence of these antibodies.